# Automated Learning Rate Scheduler for Large-batch Training

**Chiheon Kim**                                         CHIHEON.KIM@KAKAOBRAIN.COM
*Kakao Brain, South Korea*

**Saehoon Kim**                                         SHKIM@KAKAOBRAIN.COM
*Kakao Brain, South Korea*

**Jongmin Kim**                                         JMKIM@KAKAOBRAIN.COM
*Kakao Brain, South Korea*

**Donghoon Lee**                                         DHLEE@KAKAOBRAIN.COM
*Kakao Brain, South Korea*

**Sungwoong Kim**                                         SWKIM@KAKAOBRAIN.COM
*Kakao Brain, South Korea*

## Abstract

Large-batch training has been essential in leveraging large-scale datasets and models in deep learning. While it is computationally beneficial to use large batch sizes, it often requires a specially designed learning rate (LR) schedule to achieve a comparable level of performance as in smaller batch training. Especially, when the number of training epochs is constrained, the use of a large LR and a warmup strategy is critical in the final performance of large-batch training due to the reduced number of updating steps. In this work, we propose an automated LR scheduling algorithm which is effective for neural network training with a large batch size under the given epoch budget. In specific, the whole schedule consists of two phases: adaptive warmup and predefined decay, where the LR is increased until the training loss no longer decreases and decreased to zero until the end of training. Here, whether the training loss has reached the minimum value is robustly checked with Gaussian process smoothing in an online manner with a low computational burden. Coupled with adaptive stochastic optimizers such as AdamP and LAMB, the proposed scheduler successfully adjusts the LRs without cumbersome hyperparameter tuning and achieves comparable or better performances than tuned baselines on various image classification benchmarks and architectures with a wide range of batch sizes.

## 1. Introduction

In modern deep learning tasks, scaling up both the sizes of dataset and model has shown promising improvements. However, a large number of samples and slow gradient computations lead to considerably longer training time, and therefore the use of large batch size in stochastic optimization with multiple computational nodes has gain popularity to speed up for such a large-scale training. It is desirable for the optimization algorithm with the increased batch sizes to maintain the performances without increasing the amount of processed samples, namely the number of training epochs.

Recently, several works have been proposed for successful large-batch training with stochastic gradient descent (SGD), and most of them achieve it by specially designed learning rate (LR) schedules and especially LR scaling (Krizhevsky, 2014; Goyal et al., 2017; Hoffer et al., 2017). For instance, linear (Krizhevsky, 2014; Goyal et al., 2017) or square-

root (Hoffer et al., 2017) LR scaling rules according to the batch sizes effectively alleviate the performance degradation in large-batch training by compensating the reduced number of optimization steps. In addition, gradual LR warmup heuristic (Goyal et al., 2017) reduces the instability caused by large LRs and has become a standard for large-batch training. More recently, layer-wise adaptive LR scalings have been proposed to further increase the batch size and are applied to diverse tasks (You et al., 2017, 2020; Ginsburg et al., 2019). However, these LR scaling and warmup heuristics are sensitive to its hyperparameters including the LR and the warmup schedule, hence require intensive tuning effort.

There are a number of studies attempting to automate the LR scheduling to reduce such cumbersome tuning and to enhance the performance. Online or offline LR search algorithms using Bayesian optimization (Picheny et al., 2020) or meta-optimization (Schraudolph, 1999; Chen et al., 2017; Baydin et al., 2018; Donini et al., 2020) have been proposed. However, they have been limited to small (surrogate) tasks due to complexity and the feasibility of them in large-batch training is unknown.

We propose an automated online LR scheduler for large-batch training. The schedule consists of the warmup phase, where the LR is increased until the training loss no longer decreases, and the decay phase that decreases the LR. To robustly decide the phase transition on the fly, we employ Gaussian process (GP) smoothing (Rasmussen et al., 2006). This GP-based online detection is robust to small bumps and has a low computational burden. In addition, to cover a wide range of possible LR values while ensuring stability, the LR is exponentially increased from a very small value ($10^{-5}$) to a very large value (1.0) up to the half of the given epoch budget. As a result, the proposed LR scheduler can efficiently and automatically figure out not only the initial and peak LRs but also the warmup length in a data-driven way as the training progresses. From the perspective of having the warmup and the decay phase, SALSA (Zhang et al., 2020) is most similar to this work. However, SALSA relies on the backtracking line search for the warmup phase which can be burdensome, and it was not evaluated on large batch sizes.

We empirically demonstrate that the proposed LR scheduler together with an adaptive optimizer such as AdamP (Heo et al., 2021) or LAMB (You et al., 2020) achieves comparable or better performance compared to a tuned baseline in case of large batch size on various image classification benchmarks and architectures.

## 2. Methods

We propose an automated LR scheduler, called AutoWU (stands for automated warmup), which consists of two phases *warmup* and *decay*. Figure 1 describes how AutoWU works.

### 2.1 Warmup Phase

During the warmup phase, the LR starts from a very small value and is increased following an exponential schedule. It is tested at the end of each epoch whether the loss has hit the minimum value as training progresses. If the minimum has been detected, then the LR follows the predefined decay schedule. This scheme combines the warmup strategy which enhances the stability in the early phase of training with the automated LR selection.

#### 2.1.1 Warmup by exponential schedule

The proposed exponential schedule in the warmup phase has three hyperparameters: the initial LR $\eta_{\min}$, the maximum possible LR $\eta_{\max}$, and the maximum possible fraction $\rho_w \in$

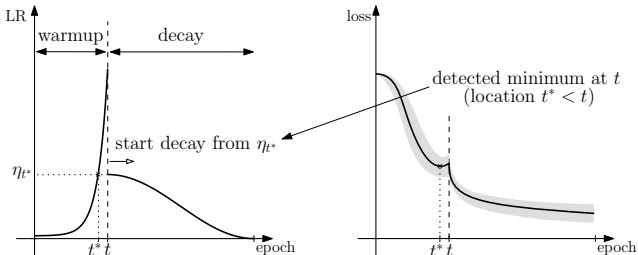

Figure 1: Conceptual diagram of AUTOWU. It adaptively switches from the warmup phase to a decay schedule (*e.g.* cosine) at $t$, upon detection of minimum loss at $t^*$ preceding $t$ with high probability. The start LR of the decay phase is also adaptively set as $\eta_{t^*}$ corresponding to $t^*$. The shaded region in the right plot indicates the variance of the observed losses, and the bold curve corresponds to the smoothed loss via GP.

$(0, 1)$ of warmup steps. Given the total number of steps $T$, the warmup schedule is defined as $\eta_0 = \eta_{\min}$ and

$$\eta_t = \gamma \eta_{t-1} \quad \text{for} \quad t \in \{1, ..., \lfloor \rho_w T \rfloor\}, \quad \text{where} \quad \gamma = (\eta_{\max}/\eta_{\min})^{1/\lfloor \rho_w T \rfloor}. \tag{1}$$

Here, if the training loss keeps decreasing, then the warmup phase can last for $\lfloor \rho_w T \rfloor$ steps, and $\eta_t$ can grow up to $\eta_{\max}$.

In comparison to the other growth rates such as the linear growth rate, this exponential growth rate enables stable and fine-grained LR exploration, especially in the early stage for large-batch training. We set $\eta_{\min} = 10^{-5}$ and $\eta_{\max} = 1$ to ensure that LRs can sweep a wide enough range of values.[1] Moreover, we set $\rho_w = 0.5$ to ensure that the LR does not grow too fast. Results regarding the sensitivity of AUTOWU with respect to $\rho_w$ and the comparison with the linear growth are found in Appendix C.2.

### 2.1.2 ONLINE MINIMUM DETECTION VIA GAUSSIAN PROCESS

A loss trajectory may exhibit local and global fluctuations due to the stochastic nature of a mini-batch computation and a highly non-convex loss landscape. This makes it difficult to robustly detect whether a loss is no longer decreasing or not in an online manner. As a remedy, we propose to use GP regression to smooth the loss curve and then conduct a decision test based on the predictive distribution.

Suppose that we have been observed (noisy) loss values $L_0, \cdots, L_{t-1}$ at step $t$. We model the loss curve $s \mapsto L_s$ (defined for $s \in [0, t]$) by a GP with a homoskedastic noise:

$$L_s = f(s/t) + \epsilon, \quad \text{where} \quad f \sim \mathcal{GP}(\theta, K_{\ell, \sigma_f}(\cdot, \cdot)) \text{ and } \epsilon \sim \mathcal{N}(0, \sigma_n^2), \tag{2}$$

where $\theta$, $K_{\ell, \sigma_f}$ and $\sigma_n^2$ denote the mean, the covariance kernel, and the noise level, respectively. When conditioned on the observed sequence $L_0, \cdots, L_{t-1}$, the predictive distribution of $f$ is again a GP and can be computed straightforwardly.[2]

We are interested in whether $f$ is no longer decreasing, *i.e.*, there exists $x^* < 1$ such that $f(x^*) < f(1)$. Hence, it is natural to compute the probability

$$\mathbb{P}_f \left( f(x) < f(1) \text{ for some } x \in [0, 1] \right), \tag{3}$$

---

1. This range of LRs is sufficient for AdamP and LAMB, but may require a modification for other optimizers, e.g., a good value of $\eta_{\max}$ for SGD would be larger.
2. See Rasmussen et al. (2006) for a good exposition on the subject.

and decide to end the warmup phase if (3) is large. However, since (3) is not easy to compute, we compute the following lower bound instead:

$$P_{\min}(f) := \max_{x \in S} \mathbb{P}_f(f(x) < f(1)), \tag{4}$$

which holds for any $S \subseteq [0, 1]$. Here, $S$ is chosen to be 500 equally-spaced points in $[0, 1]$.

**GP model details.** We use the GP model with the constant mean $\theta$ and the squared-exponential kernel $K_{\ell, \sigma_f}(x, x') = \sigma_f^2 \cdot e^{-|x-x'|^2/(2\ell^2)}$, where the length-scale $\ell$ is fixed to 0.2 to prevent model overfitting to local variations, and all other parameters $\theta$, $\sigma_f$, and $\sigma_n$ are fitted by Adam (Kingma and Ba, 2014) with LR 0.01 for 100 steps.

**Test details.** There are three main hyperparameters involved in the test: $n_{test}$, confidence $c$, and patience $p$. We used $n_{test} = 5$, $c = 0.95$ and $p = 3$ in all experiments.

Suppose that the loss trajectory $\mathcal{C} = \{(0, L_0), \cdots, (t-1, L_{t-1})\}$ has been observed at step $t$. To conduct the test, we first choose a random subset $\mathcal{C}_0 \subseteq \mathcal{C}$ of maximum 100 samples and fit the GP parameters with respect to the marginal log-likelihood of $\mathcal{C}_0$. Then, we further sample $n_{test}$ random subsets $\mathcal{C}_1, \cdots, \mathcal{C}_{n_{test}}$ of $\mathcal{C}$, each of maximum 500 samples, and infer $f_i$ conditioned on $\mathcal{C}_i$ for $i = 1, \cdots, n_{test}$. Finally, we compute $P_{\min}(f_i)$ in Eqn. (4) and if the majority of $P_{\min}(f_i)$'s exceed the confidence $c$, then it is regarded as that the minimum has been detected.

We start the decay phase only if the minimum has been detected in $p$ consecutive tests. This is to prevent premature ending of warmup due to random spikes in the loss trajectory, which was observed more often in large-batch setting in our preliminary experiments. We remark that larger $p$ implies the switch happening later; this may result in a too large LR. To remove the impact of the choice of $p$, we set the starting LR of the decay phase to be $\eta_{t^*} = \gamma^{t^*} \eta_0$ where

$$t^* := t \cdot \mathbb{E}_i \left[ \arg \min_{x \in [0,1]} \mathbb{E}_{f_i} \left[ f_i(x) \right] \right]. \tag{5}$$

The overall algorithm is summarized in Appendix A (Algorithm 1).

**Computation time.** Time complexity of exact GP inference increases cubically with the number of samples. Nonetheless, those computations are not very burdensome since they are only conducted once per epoch in this work, and we use subsampling so that the number of samples is kept constant regardless of the task (particularly the batch size). Moreover, we use GPyTorch (Gardner et al., 2018) for efficient GP computations on GPUs. Time overhead per test (fitting and 5 inferences) was less than a second in average, which is typically much smaller than the overall gradient computation time per epoch.

## 2.2 Decay Phase

In the decay phase, AutoWU follows an LR schedule with the predefined shape, but whose starting LR is adaptively determined in the warmup phase. To remove any sophistication in evaluation, we only consider two simple types of schedule in the decay phase: *cosine* (Loshchilov and Hutter, 2017) or *constant-then-cosine*. In case of cosine decay, the LR starts with the value determined by the warmup phase and is annealed toward zero with cosine schedule. This schedule is particularly appealing since it does not introduce any additional hyperparameters to consider. On the other hand, constant-then-cosine decay maintains LR constant until a predetermined fraction of epochs is left and then follows

Table 1: Comparison of test/val accuracies (%) between the baseline schedule and AUTOWU with two decay schedules (cosine and constant-then-cosine) when used with AdamP. We report the mean and the standard deviation (written in parenthesis) of three independent runs with different random seeds in case of CIFAR tasks.

| Dataset (Architecture) | Schedule | Batch size | | | |
|---|---|---|---|---|---|
| | | 256 | 1K | 8K | 16K |
| CIFAR-10 (ResNet-18) | Baseline | **96.58** (0.07) | **96.48** (0.02) | **96.05** (0.15) | 94.63 (0.06) |
| | AUTOWU + const-cos | 96.26 (0.12) | 96.20 (0.03) | 95.92 (0.22) | **94.80** (0.17) |
| | AUTOWU + cos | 96.43 (0.02) | 96.42 (0.05) | 95.77 (0.01) | 94.03 (0.26) |
| CIFAR-100 (Wide-ResNet28-10) | Baseline | 83.36 (0.38) | 83.13 (0.14) | 81.08 (0.33) | 77.62 (0.36) |
| | AUTOWU + const-cos | 83.36 (0.21) | 83.21 (0.19) | **82.32** (0.42) | **81.42** (0.35) |
| | AUTOWU + cos | **83.59** (0.46) | **83.39** (0.20) | 82.26 (0.60) | 80.25 (0.36) |
| | | 1K | 4K | 16K | 32K |
| ImageNet (ResNet-50) | Baseline | 76.28 | 76.10 | 75.02 | 74.11 |
| | AUTOWU + const-cos | **76.31** | **76.33** | **75.62** | **74.84** |
| | AUTOWU + cos | 76.19 | 75.70 | 75.22 | 74.40 |

cosine schedule. We set the fraction 0.2 (*i.e.* cosine decay in the last 20% of epochs) in all experiments. We have empirically observed that an automated LR decay based on the convergence test such as SASA+ (Zhang et al., 2020) was not decisively superior than these schedules. This probably implies that a warmup strategy including the peak LR is more critical for the final performance, especially in large-batch training setting.

## 3. Experiments

We evaluate our algorithm on image classification tasks with three benchmark datasets: CIFAR-10, CIFAR-100 (Krizhevsky et al., 2009) and ImageNet (Deng et al., 2009). We mainly consider convolutional networks: ResNet-18, Wide-ResNet-28-10 (Zagoruyko and Komodakis, 2016) for CIFAR-10 and CIFAR-100 respectively, and ResNet-50 (He et al., 2016) for ImageNet. We also conduct evaluations on ImageNet with EfficientNet-B0 (Tan and Le, 2019) and a vision transformer ViT-S/16 (Dosovitskiy et al., 2020), and the results by these other architectures are included in Appendix B.

AUTOWU only requires to know how the loss values are changing over the course of training, hence it is capable to be used with any stochastic descent algorithm. In our evaluations, the state-of-the-art Adam-based optimizer, AdamP (Heo et al., 2021) is used as the base optimizer due to their better stability than SGD and its variants with large batch sizes.[3] The results when coupled with the layer-wise adaptive optimizer, LAMB (You et al., 2020), are also presented in Appendix B.

The proposed AUTOWU scheduler is compared with the conventional baseline LR schedule on each task with multiple batch sizes: $\{256, 1024, 8192, 16384\}$ for CIFAR (200 epochs) and $\{1024, 4096, 16384, 32768\}$ for ImageNet (120 epochs). The baseline LR schedule consists of 5 epochs of a linear warmup from 0 to a predetermined peak LR followed by a cosine decay to 0. Here, following a common LR scaling practice for large-batch training, the peak LR is scaled with the batch-size according to the square-root scaling rule, specifically, $\eta_{base}\sqrt{B/256}$ for batch size $B$. We set $\eta_{base} = 0.001$ in all experiments by empirically tuning it as the common base LR that is well performed across tasks and architectures. We verify that the base LR as well as the warmup length strongly affect the performances of the baseline LR scheduler for large-batch training (see Appendix C.1), however the proposed AUTOWU scheduler removes these tuning efforts.

---

3. In our preliminary experiments, we have observed that instability of SGD could be mitigated with gradient clipping but extensive tuning effort was required to match the same level of performances.

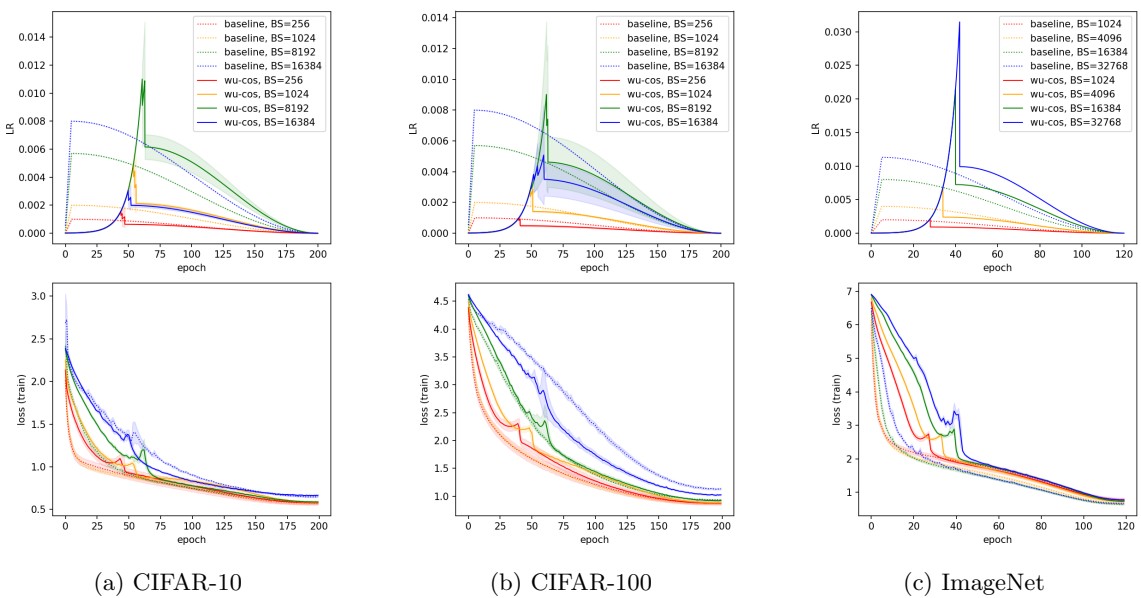

(a) CIFAR-10          (b) CIFAR-100          (c) ImageNet

Figure 2: LR schedules (top row) and training loss curves (bottom row) of the baseline and AUTOWU with cosine decay (denoted wu-cos in the legend) on CIFAR and ImageNet tasks. Shadowed area represents the standard deviation computed over different seeds.

As shown in Table 1, the proposed AUTOWU performs on par with or better than the baseline LR schedule across diverse batch sizes and tasks. Especially, with large batch sizes of 8K and 16K for CIFAR-100 and 16K and 32K for ImageNet, AUTOWU with constant-then-cosine decay significantly reduces the performance drops in comparison to the conventional baseline. In addition, overall, the constant-then-cosine decay is slightly better than the cosine decay when combined with AUTOWU which means that as we find the safe peak LR automatically for a long warmup time, retaining the found LR longer would be better.

Figure 2 shows the LR schedule that AUTOWU found and the training loss curve in each task. For any batch size and task, AUTOWU robustly detects the minimum loss in an online manner by GP. Also, interestingly, the starting LR of the decay phase by AUTOWU grows as the batch size increases, and the rate is similar to the baseline scaling of the peak LR even though the warmup epochs are quite different. Actually, the found combination of the peak LR and the warmup length is similar to the search results on the baseline schedule as shown in Appendix C.1. More ablations on AUTOWU are presented in Appendix C.

## 4. Conclusion

In this work, an automated LR scheduler especially for fast training using large batch sizes is proposed. The proposed LR schedule has the warmup phase followed by the decay phase, where the LR is exponentially increased until the training loss no longer decreases and then decreased to zero until the end of training. The online detection of the minimum loss has been efficiently and robustly realized by GP regression. Empirical evaluation demonstrates that our automated LR scheduler appropriately adapts the whole warmup procedure as well as LRs for any batch size and task, and consequently results in comparable or better performances in comparison to the fine-tuned LR schedulers. Our implementation will be available at https://github.com/kakaobrain/autowu.

## Acknowledgments

We would like to acknowledge and thank Woonhyuk Baek and Brain Cloud Team at Kakao Brain for their support.

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

---

**Algorithm 1** Automated LR scheduler (AUTOWU)

---

**Hyperparameters:** $\eta_{\min}$, $\eta_{\max}$, $\rho_w \in (0, 1)$, $n_{test}$, $c$ (confidence), $p$ (patience)
**Inputs:** $\theta_0$, $\mathcal{A}$ (optimizer), $T$ (the number of total steps)
$\eta_0 \leftarrow \eta_{\min}$, $\gamma \leftarrow (\eta_{\max}/\eta_{\min})^{1/\lfloor \rho_w T \rfloor}$, patience_flag $\leftarrow 0$, $\mathcal{C} \leftarrow \emptyset$
**for** $t$ **from** 0 **to** $T-1$ **do**
    Sample $\xi_t$.
    Compute loss and stochastic gradient: $L_t = L(\theta_t; \xi_t)$, $g_t = \nabla_{\theta_t} L_t$.
    Take an optimizer step of $\mathcal{A}$ w.r.t. LR $\eta_t$ and $g_t$ to compute $\theta_{t+1}$.
    **if** warmup phase **then**
        $\mathcal{C} \leftarrow \mathcal{C} \cup \{(t, L_t)\}$.
        **if** end of epoch **then**
            Subsample $\mathcal{C}_0, \cdots, \mathcal{C}_{n_{test}}$ from $\mathcal{C}$.
            Fit GP parameters w.r.t. marginal log-likelihood of $\mathcal{C}_0$.
            Infer $f_i$ conditioned on $\mathcal{C}_i$ for $i = 1, \cdots, n_{test}$.
            Compute $P_{\min}(f_i)$ as in Eqn. (4).
            **if** $\#(i : P_{\min}(f_i) > c) > n_{test}/2$ **then**
                patience_flag $\leftarrow$ patience_flag $+ 1$
            **else**
                patience_flag $\leftarrow 0$
            **end if**
            **if** patience_flag $\geq p$ **then**
                Compute $t^*$ according to Eqn. (5).
                Set $\eta_{t+1} \leftarrow \gamma^{t^*} \eta_0$ and switch to decay phase.
            **end if**
        **end if**
        Update LR if not switched: $\eta_{t+1} \leftarrow \gamma \eta_t$.
    **else**
        Compute $\eta_{t+1}$ according to cosine or cosine-then-decay schedule.
    **end if**
**end for**

---

## Appendix A. More Details

The overall algorithm of AUTOWU is depicted in Algorithm 1. All implementations are based on PyTorch 1.7.0 and GPyTorch 1.3.0. Experiments are conducted on NVIDIA Tesla V100 GPUs with 32GB memory.

We use the weight decay of 0.1, $\beta_1 = 0.9$, $\beta_2 = 0.999$, $\epsilon = 10^{-8}$ and $\delta = 0.1$ for AdamP and the weight decay of 0.1, $\beta_1 = 0.9$, $\beta_2 = 0.999$ and $\epsilon = 10^{-6}$ for LAMB.

In all experiments involving CIFAR-10 or CIFAR-100, the corresponding network is trained with AutoAugment (Cubuk et al., 2018), Cutout (DeVries and Taylor, 2017), and label smoothing (Müller et al., 2019) with factor 0.1 for 200 epochs. We report the mean and the standard deviation from three independent runs with different random seeds. In case of ResNet-50 training on ImageNet, the standard set of augmentations as in He et al. (2016) is used for ImageNet and the network is trained for 120 epochs. We report the result of a single run with a fixed random seed in all ImageNet experiments due to resource constraints.

## Appendix B. Further Evaluations

In this section, further evaluation results which were not discussed in Section 3 are presented.

Table 2: Comparison of test accuracies (%) on CIFAR-10 and CIFAR-100 between a default schedule and AutoWU with LAMB.

| Dataset (Architecture) | Schedule | Batch size | | | |
|---|---|---|---|---|---|
| | | 256 | 1K | 8K | 16K |
| CIFAR-10 (ResNet-18) | Baseline | **96.37** (0.16) | **96.39** (0.14) | **95.92** (0.09) | **95.36** (0.08) |
| | AutoWU + const-cos | 96.26 (0.09) | 96.17 (0.07) | 95.53 (0.13) | 94.47 (0.23) |
| | AutoWU + cos | 96.23 (0.14) | 96.14 (0.09) | 95.37 (0.10) | 94.19 (0.22) |
| CIFAR-100 (WideResNet28-10) | Baseline | **83.38** (0.15) | **83.61** (0.01) | 82.16 (0.19) | **79.99** (0.30) |
| | AutoWU + const-cos | 82.86 (0.16) | 83.11 (0.32) | **82.44** (0.11) | 79.36 (0.47) |
| | AutoWU + cos | 82.92 (0.33) | 82.88 (0.38) | 81.31 (0.06) | 77.57 (0.50) |

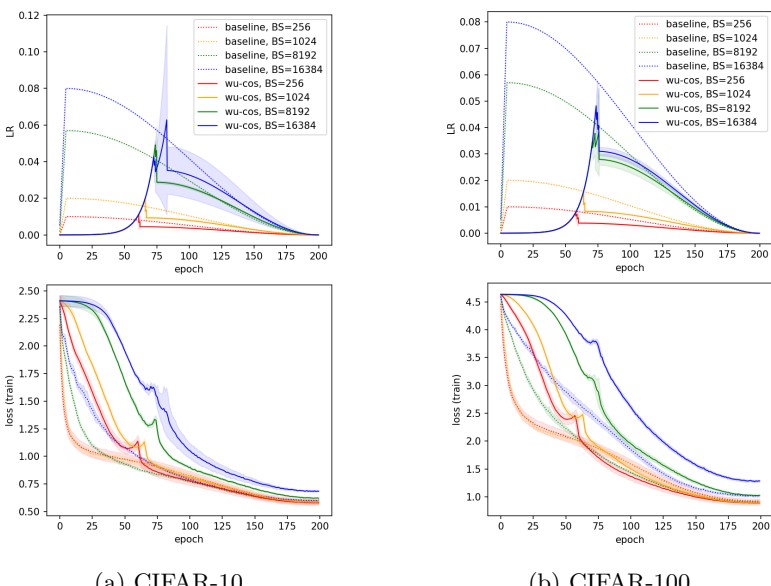

(a) CIFAR-10          (b) CIFAR-100

Figure 3: LR schedules (top row) and training loss curves (bottom row) of the baseline and AutoWU with cosine decay on CIFAR tasks, when the base optimizer is LAMB.

Firstly, the comparison between AutoWU and the baseline schedule when coupled with LAMB is made. The peak LR of the baseline is scaled as $\eta_{base}\sqrt{B/256}$ for batch size $B$. The base LR $\eta_{base}$ is chosen 0.01 rather than 0.001, since the latter has resulted in worse performances. The results are summarized in Table 2, and the plot of LR schedules and training loss curves can be found in Figure 3. We observe that general tendency is similar to that of AutoWU coupled with AdamP, but the resulting performances are slightly worse than those obtained by the baseline schedule with an exception of batch size 8192 on CIFAR-100. However, we emphasize that AutoWU does not require tuning, in contrast to the fact that LAMB can be more sensitive to $\eta_{base}$ (Chen et al., 2021) and adjusting $\eta_{base}$ to 0.01 in baselines was necessary to achieve the reported performance.

We also evaluate the performance of AutoWU on ImageNet classification task with two other architectures: EfficientNet-B0 and ViT-S/16. The training configuration of EfficientNet-B0 is identical to that of ResNet-50, except that the label smoothing of factor 0.1 is used. For ViT-S/16, it is trained for 300 epochs with augmentations as described in Touvron et al. (2020). The peak LR of the baseline is scaled as $\eta_{base}\sqrt{B/256}$ with $\eta_{base} = 0.001$ for EfficientNet-B0 and $\eta_{base} = 0.0005$ for ViT-S/16. Results are found in

Table 3: Comparison of top-1 validation accuracies (%) between the baseline schedule and AutoWU, in case of ImageNet training on EfficientNet-B0 and ViT-S/16 with AdamP.

| Dataset (Architecture) | Schedule | Batch size | | | |
|---|---|---|---|---|---|
| | | 1K | 4K | 16K | 32K |
| ImageNet (EfficientNet-B0) | Baseline | **75.03** | **76.00** | 74.23 | **75.17** |
| | AutoWU + const-cos | 74.58 | 75.43 | 75.01 | 73.99 |
| | AutoWU + cos | 74.90 | 75.81 | **75.44** | 74.34 |
| ImageNet (ViT-S/16) | Baseline | **79.37** | **79.34** | 77.39 | **73.92** |
| | AutoWU + const-cos | 77.65 | 78.14 | 75.67 | 72.73 |
| | AutoWU + cos | 79.01 | 79.15 | 76.95 | 72.38 |

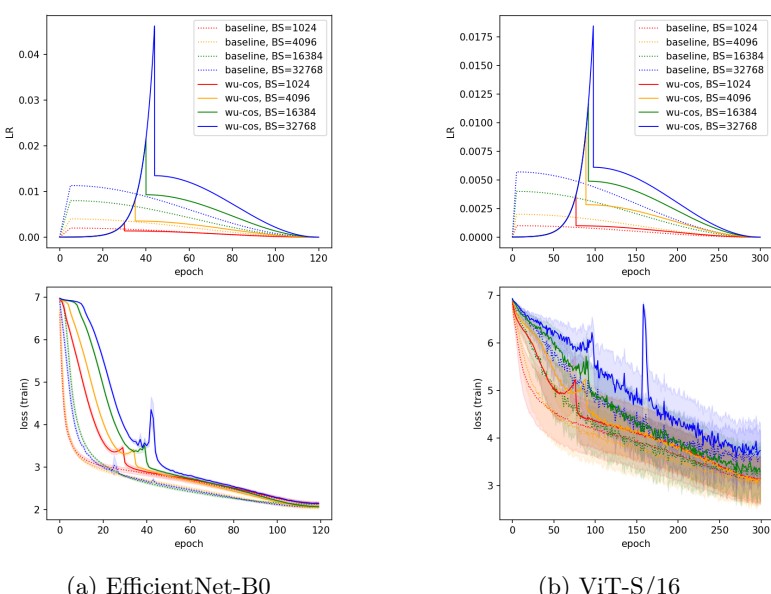

(a) EfficientNet-B0        (b) ViT-S/16

Figure 4: LR schedules (top row) and training loss curves (bottom row) of the baseline and AutoWU with cosine decay on ImageNet with EfficientNet-B0 and ViT-S/16, when the base optimizer is AdamP.

Table 3, and the plots of LR schedules and training loss curves are found in Figure 4. Even though the performances are a little bit lower compared to the tuned baseline, AutoWU stably works on other architectures without any specific hyperparameter tuning.

## Appendix C. Ablation Studies

### C.1 Sensitivity of the baseline with respect to the warmup schedule

We compare the performances of the baseline LR scheduling according to various numbers of warmup epochs and peak LRs. Namely, the LR is linearly increased from 0 to the peak LR $\eta_{peak}$ for $n_{warmup}$ epochs then decayed to 0 via cosine schedule, and the experiment is carried out for $\eta_{peak} \in \{0.002, 0.004, 0.008, 0.016, 0.032\}$ and $n_{warmup} \in \{5, 20, 40, 60\}$. Here, we use ResNet-50 trained by AdamP with a batch size of 16384 on ImageNet, and therefore the configuration $(\eta_{peak}, n_{warmup}) = (0.008, 5)$ corresponds to the baseline schedule (presented in Table 1).

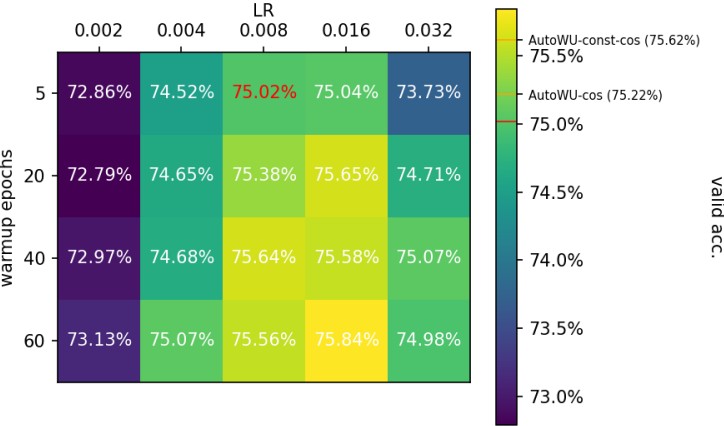

Figure 5: Comparison of validation accuracies over the choices to the peak LR $\eta_{peak}$ and the number of warmup epochs $n_{warmup}$, in case of ResNet-50 training on ImageNet with AdamP and a batch size of 16384. The schedule presented as the baseline in Table 1 corresponds to $(\eta_{peak}, n_{warmup}) = (0.008, 5)$ and colored red in the plot. Additionally, the performances of AUTOWU with cosine and constant-then-cosine decay are annotated as orange lines in the colorbar on the right.

Figure 5 demonstrates how the performance of ResNet-50 on ImageNet changes with respect to $\eta_{peak}$ and $n_{warmup}$. The value of optimal LR is either 0.008 or 0.016 when $n_{warmup}$ is fixed, which includes our baseline. On the other hand, we find that the validation accuracy is improved as $n_{warmup}$ increases and the best performance is 75.84% obtained by $(\eta_{peak}, n_{warmup}) = (0.016, 60)$. We remark that not only the performances of AUTOWU with cosine and constant-then-cosine decay (75.22% and 75.62%, respectively) are fairly close to the best performance but also the peak LR and the warmup epochs found by AUTOWU, $(0.0215, 40)$ as shown in Figure 2(c), are similar to the best configuration on the baseline schedule. This validates the effectiveness of the proposed algorithm.

## C.2 Dependency of AUTOWU on the warmup schedule

We have argued that the exponential growth in the warmup phase stabilizes the training and enables a fine-grained LR exploration. To demonstrate this, we compare the linear growth and the exponential growth in case of ResNet-50 training on ImageNet with AdamP and batch size 16384. Specifically, we consider the linear schedule in the warmup phase defined as

$$\eta_t = \eta_{\min} + (\eta_{\max} - \eta_{\min}) \cdot \frac{t}{\lfloor \rho_w T \rfloor} \quad \text{for} \quad t \in \{0, \cdots, \lfloor \rho_w T \rfloor\}, \tag{6}$$

where $\eta_{\min} = 10^{-5}$ and $\eta_{\max} \in \{0.1, 1.0\}$. In both choices of $\eta_{\max}$, the cosine schedule is used in the decay phase and all hyperparameters are set identical as in case of AUTOWU with exponential warmup schedule.

Figure 6 shows the LR schedule, the training loss and the validation accuracy of the baseline, AUTOWU with exponential growth, and AUTOWU with linear growth as described above. In terms of the validation accuracy, the baseline and AUTOWU with exponential growth achieves 75.02% and 75.22% respectively, while AUTOWU with linear growth

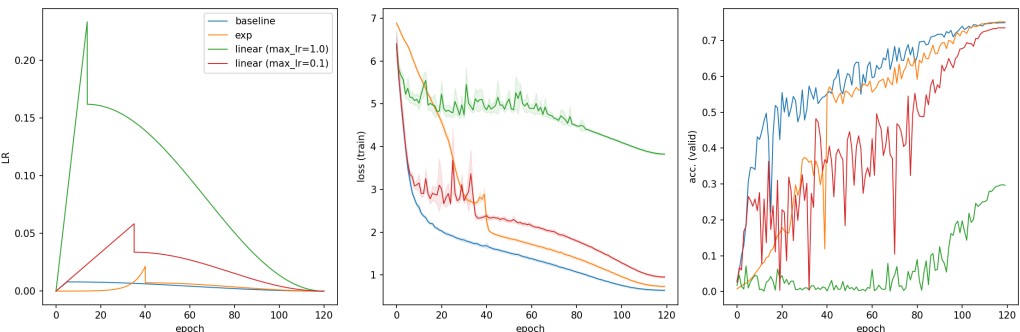

Figure 6: Comparison of the linear growth and the exponential growth in the warmup phase of AutoWU. ResNet-50 is trained on ImageNet with a batch size of 16384, AdamP optimizer, and the corresponding scheduler.

Table 4: Sensitivity of AutoWU (+cos) with respect to the maximum fraction of warmup ($\rho_w$).

| $\rho_w$ | Batch size | | | |
|---|---|---|---|---|
| | 1K | 4K | 16K | 32K |
| 0.125 | 75.89 | 75.59 | 74.52 | 73.40 |
| 0.25 | **76.60** | **76.04** | **75.28** | 73.89 |
| 0.5 | 76.19 | 75.70 | 75.22 | **74.40** |

achieves 29.65% for $\eta_{\max} = 1.0$ and 73.56% for $\eta_{\max} = 0.1$. This implies that the linear warmup makes AutoWU very sensitive to the choice of $\eta_{\max}$, hence a significant amount of effort to tune $\eta_{\max}$ must be made, obliterating the purpose of automated LR scheduling.

The growth factor $\gamma$ in Eqn. 1 in AutoWU with the exponential growth is completely determined by $\rho_w$ when $\eta_{\min}$ and $\eta_{\max}$ are set to the sufficiently small and large values; $\gamma$ is increased if $\rho_w$ is decreased. Therefore, we also compare the performances of AutoWU with the cosine decay for different values of $\rho_w$.

Specifically, ResNet-50 is trained on ImageNet with a batch size of 16384, AdamP, and AutoWU for $\rho_w \in \{0.125, 0.25, 0.5\}$, and the results are summarized in Table 4. Here, $\rho_w = 0.5$ corresponds to the default configuration which is also reported in Table 1. The best performances are attained by $\rho_w = 0.25$ or $\rho_w = 0.5$, and when $\rho_w = 0.125$, the performances are degraded.

We plot the relation between the starting LR $\eta_{t^*}$ of the decay phase and the choice of $\rho_w$ in Figure 7. Larger batch size or faster growth implies the bigger starting LR in the decay phase in general, but this breaks down when the batch size becomes very large. We have observed that this "critical batch size" is larger with larger $\rho_w$ (*i.e.* smaller $\gamma$), and supports the intuition that slow growth of LR in the warmup phase stabilizes the overall training dynamics. Such an observation is also consistent with the previous works (Smith and Topin, 2019; Cohen et al., 2021).

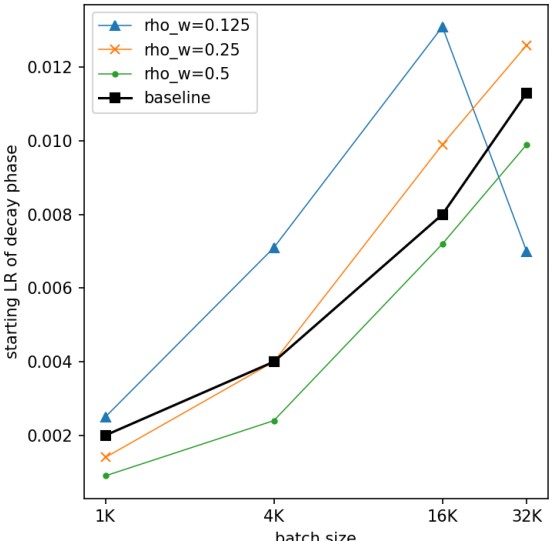

Figure 7: Dependency of AutoWU on $\rho_w$ in terms of the starting LR of the decay phase. Plotted are the starting LRs of the decay phase of AutoWU for $\rho_w \in \{0.125, 0.25, 0.5\}$ together with the peak LRs of the baseline, for batch sizes $\{1024, 4096, 16384, 32768\}$.

## Appendix D. Further discussions

We did not present the study investigating the impact of other hyperparameters including the confidence $c$, patience $p$, and the choice of decay schedule on the performance of the proposed algorithm AutoWU. While we leave a thorough study as a future work, we briefly discuss below on a few observations from our preliminary experiments.

- It seems that the choice of confidence $c$ does not matter much, as $P_{\min}(f)$ in Eqn. (4) become very close to 1 when the loss starts to increase. We also note that conducting $n_{test} > 1$ tests on sub-sampled loss curves made the test more robust.

- The choice of $p$ was more critical. Note that AutoWU is an algorithm which detects whether the loss minimum lies in the past or not; this implies that AutoWU may be susceptible to a local fluctuation of loss, especially when the number of steps is small (or equivalently, the batch size is large). To reduce such a sensitivity, the patience $p$ to wait whether the observed minimum is local or not is introduced. Utilizing a large value of $p$ makes the test more robust, but simultaneously lets LR to grow until a large value hence training may diverge. We found that setting $p = 3$ was working well for all experiments.

- We argued that SALSA (Zhang et al., 2020) is the most similar work to AutoWU. In terms of warmup scheduling, SALSA employs a line search based method to warmup the initial training dynamics and ends the warmup phase *when the loss stops to decrease*, in the similar spirit as this work. However, the extra computation required for the line search was less appealing than a simple exponential schedule, and we have observed that the line search based method in general is sensitive to the Armijo constant especially for large batch sizes.

