# OpenReview forum: "Automated Learning Rate Scheduler for Large-batch Training"
_ICML.cc/2021/Workshop/AutoML — AutoML@ICML2021 Poster_

### Official Review · Reviewer_9z4U · 2021-06-13
**Interesting work but the experiments are not fully convincing.**

**Rating:** 6
**Confidence:** 3

**Review:**

The work presents a method of automatically tuning the learning rate for large batch training.
To do so, the overall budget is divided into a "warmup" and a "decay" phase. In the warmup phase the learning rate is rapidly increased until no learning progress can be detected. The decay phase then starts from an estimated well performing learning rate (determined from observations in the warmup phase) which is decreased according to a predefined schedule.
The switching point is determined via a GP at the end of each epoch (according to Algorithm 1).
Although the elements of the proposed method are explained well it is often not clear why some design decisions were taken. In some cases (such as using an exponential growth rate in the warmup phase) the explanations are given in the appendix though for other aspects no explanations as to why a potential hyperparameter has been fixed to a particular value. Following this, I disagree with the claim that the method does not require any tuning (as stated in appendix B). The method comes with some well performing defaults which probably need be adapted to the task at hand to get the best performance, although I agree that the presented method (with its default settings) seems to perform well compared to the baseline. In appendix B examples are given where the presented method with it's defaults works well but is slightly outperformed by the baseline.
Ablation results are given in appendix C but some interesting hyperparameters such as the "patience"  or the confidence in the GPs minimum are not evaluated nor discussed in text.
Thus overall I'm not fully convinced by the experiments as they do not support that the method does not require tuning.
With the discussion on SALSA I would have also expected to see a comparison to this method to showcase that the proposed method can outperform what is described as being "[...] most similar to this work.".

Nevertheless I believe the presented work is very interesting and including it in the workshop could lead to interesting discussions. In particular I believe the "online minimum detection via GPs" and the explicit separation of training into "warmup" and "decay" phases and how information from one phase can be carried over to the other is of interest to the community. Thus my vote is tipping towards acceptance.

Minor comments:
Algorithm 1 is missing a few details which I would recommend to rectify to make it easier to follow.
Instead of ending with "Compute $\eta_{t+1}$ with respect to the phase" it would be clearer if you were to refer to the decay schedule. Also the "flag" hyperparameter was confusing to me at first and instead of calling it just "flag" it would be clearer to name it something like "patience_flag".

---

### Official Review · Reviewer_SiN1 · 2021-06-15
**Interesting scheduler (with many more hyperparameters than what declared)**

**Rating:** 6
**Confidence:** 3

**Review:**

This paper presents a learning rate scheduler with a warm-up stage and a decay stage that automatically detects the switching point between the two by modelling the learning curve with GP regression.

I found the method interesting, especially in that it strongly alleviates the burden of extensively tuning the initial learning rate.
Instead, the authors propose an exponential warm-up stage from a very small initial value.
Then, cosine annealing is used in the decaying stage.

I found the explanation on the usage of GPs to model the learning curve and detecting the point to change regime a bit confusing, also because some quantities appear not properly defined (e.g. what are the f_i? what about t in Eq (5) ). Despite this, I think the approach generally makes sense, although I would have appreciated seeing also a comparison with a simpler backtracking procedure: for instance, you could wait for divergence in the warm-up stage and then simply revert to the point that reached the minimum of the loss.

The other two main weaknesses are:
- The presence of many more hyperparameters that what declared: (e.g. choice of kernel, confidence to use, scheduler for the decay phase, and others). In fact, I am ready to believe the method is reasonably insensitive to the hyperparameters declared (as it can detect when to change), but am quite unsure about these others.
- Experiments comparisons don't look entirely fair to me. A fairer approach would be to make time-controlled experiments where you randomly sample all hyperparameters of the competing methods (as e.g. in Donini 2020)

Somewhat minor:
- There are sentences throughout the paper regarding typical optimization behaviour (e.g. ``'' This is to prevent premature ending of warmup due to random bumps in the loss trajectory, which typically occurs more often in large-batch setting.'') that need some sort of backup, either citing some work  or with some (even toy) experiment

---

### Decision · Program_Chairs · 2021-06-21

Accept (Poster)